# Existence of multi-radical and closed-shell semiconducting states in post-graphene organic Dirac materials

Isaac Alcón [1], Francesc Viñes[1], Iberio de P.R. Moreira [1] & Stefan T. Bromley [1,2]

Post-graphene organic Dirac (PGOD) materials are ordered two-dimensional networks of triply bonded $sp^2$ carbon nodes spaced by $\pi$-conjugated linkers. PGOD materials are natural chemical extensions of graphene that promise to have an enhanced range of properties and applications. Experimentally realised molecules based on two PGOD nodes exhibit a bi-stable closed-shell/multi-radical character that can be understood through competing Lewis resonance forms. Here, following the same rationale, we predict that similar states should be accessible in PGOD materials, which we confirm using accurate density functional theory calculations. Although for graphene the semimetallic state is always dominant, for PGOD materials this state becomes marginally meta-stable relative to open-shell multi-radical and/or closed-shell states that are stabilised through symmetry breaking, in line with analogous molecular systems. These latter states are semiconducting, increasing the potential use of PGOD materials as highly tuneable platforms for future organic nano-electronics and spintronics.

[1] Departament de Ciència de Materials i Química Física & Institut de Química Teòrica i Computacional (IQTCUB), Universitat de Barcelona, Carrer Martí i Franquès 1, 08028 Barcelona, Spain. [2] Institució Catalana de Recerca i Estudis Avançats (ICREA), 08010 Barcelona, Spain. Correspondence and requests for materials should be addressed to I.Aón. (email: ialcon@ub.edu) or to S.T.B. (email: s.bromley@ub.edu)

The triply bonded $sp^2$ hybridised form of carbon has become crucially significant in both applied and fundamental solid state research[1] due to its intrinsic role as a building block of some of world's most widely studied materials (e.g. carbon nanotubes[2], fullerenes[3] and graphene[4]). Organic nanosystems based on $sp^2$ carbon networks are seen by many as an essential basis for many future technologies[1], arguably provoking a paradigm shift in materials science. Graphene, the simplest two-dimensional (2D) hexagonal network of $sp^2$ hybridised carbon, plays an especially prominent role in this respect[5]. Since its isolation[4], researchers have become fascinated with its many exceptional properties, which are progressively being more fully understood and exploited in many fields of science and technology. One particularly appealing characteristic of graphene is the existence of Dirac cones in its electronic band structure[5]. This feature, due to specific bands crossing at the Fermi level, $E_F$, gives rise to semimetallicity and ultra-high electron mobilities, and has promoted graphene as a potential candidate to replace silicon in the next generation of nanoelectronic devices[5]. However, the intrinsic connection of these features with the extended high symmetry of its electronic state limits its use in applications requiring nanostructuring (e.g., nano-electronics[6]).

The discovery of graphene has triggered a global search for other 2D materials with similar characteristics, but with higher electronic tunability and greater resilience to nanostructuring. In this direction, ab initio computational modelling has predicted that Dirac cones should be in the band structures of a number of non-carbon-based[7–9] and carbon-based[10, 11] 2D materials (the latter often referred to as post-graphene organic Dirac materials (PGOD)). An important class of PGOD materials are the so-called graphynes[12, 13] and graphdiynes[13], which present $sp^2$ carbon atoms linked via alkyne groups in a variety of ways[14]. Although some graphdiynes have been synthesised[15], those predicted to be Dirac materials are still awaiting experimental realisation. Recently, a series of 2D conjugated networks were

predicted to present Dirac cones in their bandstructures[10]. These networks present a common basic skeleton: a hexagonal array of triply bonded $sp^2$ carbon atoms (or nodes of the hexagonal network) connected by $\pi$-conjugated linkers (see Fig. 1a). Replacing the trigonal $sp^2$ carbon nodes by other elements or aromatic groups opens up a finite band gap around $E_F$, highlighting the key role of these $sp^2$ carbon centres. In ref. [6]. one proposed material was formed from $sp^2$ carbon atoms connected with biphenyl linkers (graphdiphenyl–GDP). Herein, we further propose graphphenyl (GP), obtained when using phenyl ring as linker (see first two structures in Fig. 1b). Fitting this general scheme, α-graphyne and α-graphdiyne can be thought of as hexagonal networks of $sp^2$ carbon nodes connected by ethyn-1,2-diyl (-C≡C-) or buta-1,3-diyn-1,4-diyl (-C≡C-C≡C-) alkyne linkers, respectively. Finally, graphene is the densest hexagonal array of $sp^2$ carbon nodes directly linked by bonding based on $\pi$–$\pi$ orbital overlap. This family of 2D materials is summarised in Fig. 1b.

Focusing on GDP (first structure in Fig. 1b), we note that its basic structure coincides with that of Tschitschibabin's hydrocarbon (see Fig. 1c)[16], one of the oldest bi-radical compounds ever synthesised. The triply bonded methyl radical carbon atoms in Tschitschibabin's hydrocarbon bear the two unpaired electrons of the bi-radical molecule. Due to the connectivity linking these two radical sites, the molecule possesses two possible Lewis resonance forms: 1) the open-shell anti-ferromagnetic (AFM) one with two unpaired localised electrons (Fig. 1c, left) and 2) the closed-shell one with paired $\pi$-conjugated electrons forming a quinoidal-like structure (Fig. 1c, right). Experimentally, the molecule presents a mixture of both extreme resonance forms in its electronic ground state, displaying a partial open-shell bi-radical character mixed with structural quinoidal character-istics[17]. Thus, seen as a 2D extended version of Tschitschibabin's bi-radical, and considering periodically extended Lewis resonance forms of the type shown in Fig. 1c, GDP may also be expected to display analogous electronic states (i.e. AFM and quinoidal).

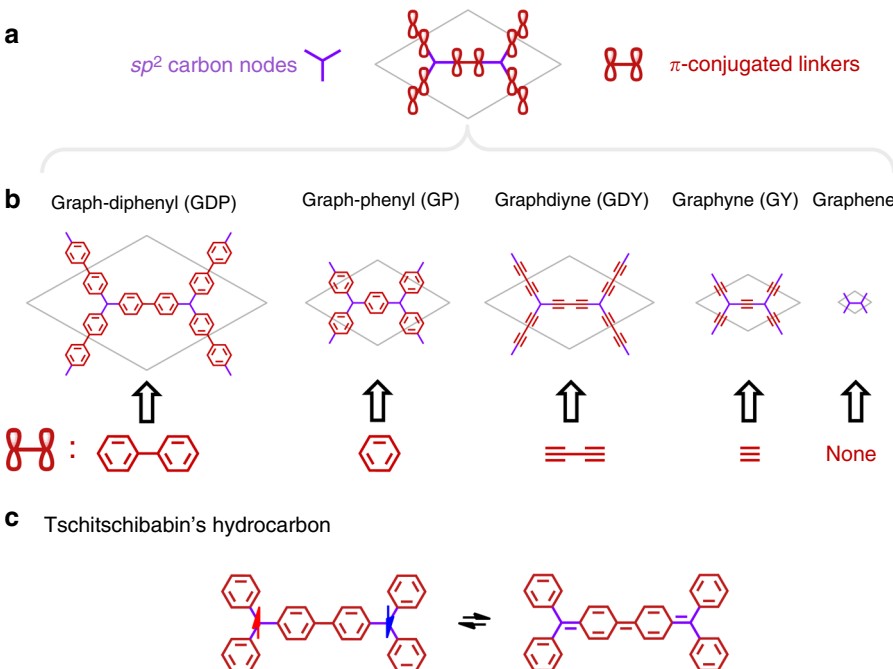

**Fig. 1** PGOD chemical structures. **a** Basic covalent skeleton composed of triply bonded $sp^2$ carbon atoms (purple) and $\pi$-conjugated linkers (red). **b** Series of graphene and four PGOD materials with different $\pi$-conjugated linkers. **c** Schematic representation of the bi-radical (left) and quinoidal (right) Lewis resonance forms stabilized by spin polarisation and structural distortion, respectively, in the Tschitschibabin's hydrocarbon. Note: Hydrogen atoms in phenyl rings are omitted for clarity

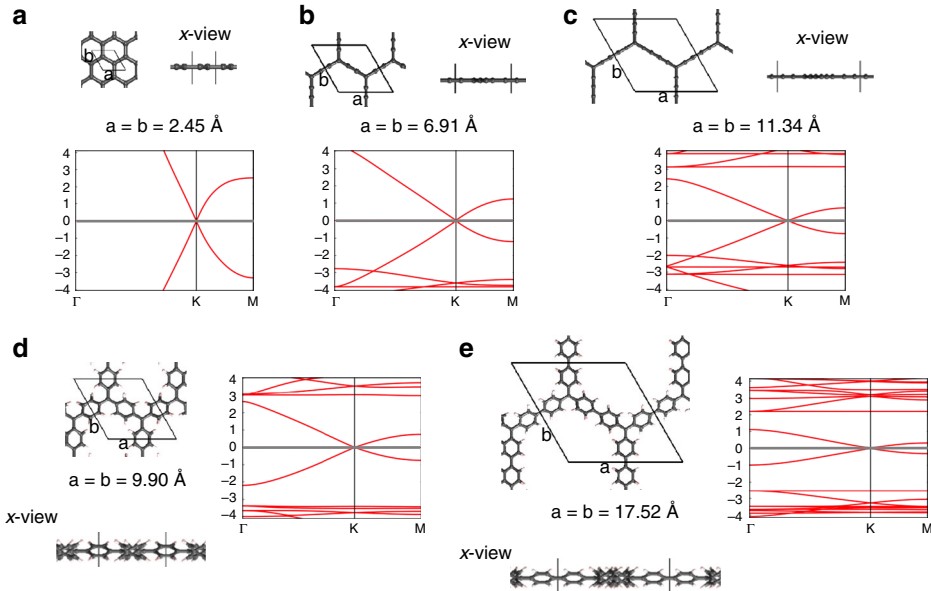

**Fig. 2** Semimetallic state. Fully optimised structures of: **a** graphene, **b** GY, **c** GDY as obtained from spin-unrestricted calculations using the PBE0 hybrid functional. The results for GP (**d**) and GDP (**e**) were obtained from single point PBE0 calculations using fully symmetrised structures. The unit cell used in each case is highlighted from both *z* and *x* views, indicating the optimised cell parameters. The corresponding electronic band structures are also provided, setting $E_F$ to zero energy in each case. Band structure energies are in eV. The broken symmetry structures for GP and GDP and their associated band structures are shown in Supplementary Fig. 4

Moreover, due to the common basic structure shared by all materials depicted in Fig. 1b (i.e. hexagonal lattice of $sp^2$ carbon nodes connected by $\pi$-conjugated linkers), it is chemically reasonable to expect that these extended resonance forms should lead to the accessibility of similar electronic states in all cases.

As far as we are aware, none of the family of materials in Fig. 1b have been reported to have intrinsic localised multi-radical and quinoidal electronic states. However, for graphene, experimental and theoretical studies have shown that such localised states can be generated by: (i) nano-patterning[18–20], (ii) cutting it into nanoribbons[18, 21, 22], (iii) covalent grafting of molecules[23] or atoms[24, 25], (iv) inclusion of defects[26], or (v) applying external strains[27, 28]. Analogously, computational modelling has predicted similar states in $\alpha$-graphyne (GY) upon application of strain[29] or by forming $\alpha$-GY nanoribbons[30, 31]. These studies lend support to our chemically motivated proposal that extended resonance forms may lead to open-shell multi-radical and/or closed-shell solutions being stabilised in all the materials depicted in Fig. 1b.

In this work, we present a systematic computational modelling study employing accurate density functional theory (DFT) based calculations to assess the existence and stability of low lying localised electronic states in all the 2D PGOD materials shown in Fig. 1b. Inspired by the limiting resonance forms of Tschitschibabin's hydrocarbon, our results demonstrate the existence of stable open-shell multi-radical and/or closed-shell quinoidal semiconducting electronic solutions in the considered PGOD materials. These results thus establish a clear link between the world of bi-radical/quinoidal molecular systems[32–35] and that of PGOD materials. In all cases, these solutions are found to be energetically accessible and to significantly affect the electronic properties of the host materials, highlighting the potential of such systems for future organic electronics/spintronics applications.

## Results

**Choice of appropriate DFT functional.** Largely due to their computational efficiency and relative accuracy, DFT based

approaches have become the dominant method of choice when simulating the electronic structures of periodic materials. For numerous cases (e.g. metallic systems, mechanical properties) the standard generalised gradient approximation (GGA) produces reliable results. However, hybrid functionals (i.e. functionals which incorporate a fraction of Hartree–Fock-like exchange (HFE)) are now being widely adopted by material's modellers[36], especially for capturing the most subtle electronic features displayed by many systems (e.g. magnetism[37], mixed valence compounds[38], semiconductors[39]). The molecular computational chemistry community has a long tradition of using hybrid functionals where, especially for electronic structure, they are known to generally outperform GGA functionals[40, 41]. A particular case is that of bi-radical/quinoidal compounds (such as Tschitschibabin's hydrocarbon), which have attracted much attention in recent years for its use in organic electronics[33] and organic magnetism[34, 42]. When these compounds are reduced or doped by other elements (e.g., N), charge transfer phenomena between the two $sp^2$ centres takes place[43, 44]. The computational assessment of the electronic ground state of such molecular systems is thus quite challenging[38]. DFT calculations employing GGA functionals over-delocalise valence electrons, and cannot stabilise the localised states found in experimental observations. Calculations using hybrid functionals, on the other hand, are found to be able to capture the quite complex and sensitive electronic structure of such $\pi$-conjugated mixed valence compounds[38, 45].

Most of the periodic modelling studies assessing the electronic structure of Dirac materials are performed using pure GGA[9, 12, 46], or local density approximation (LDA)[10, 47] functionals, and seldom utilise hybrid functionals[8, 27]. A reason for this might be that the high tendency of GGA to delocalise electrons ensures the success in finding the known, and highly studied, semimetallic solutions. However, knowing how critical the proportion of HFE is for obtaining the correct localised solutions in $\pi$-conjugated molecular systems[38, 45], it is reasonable to think that possible localised solutions in Dirac cone materials might have remained

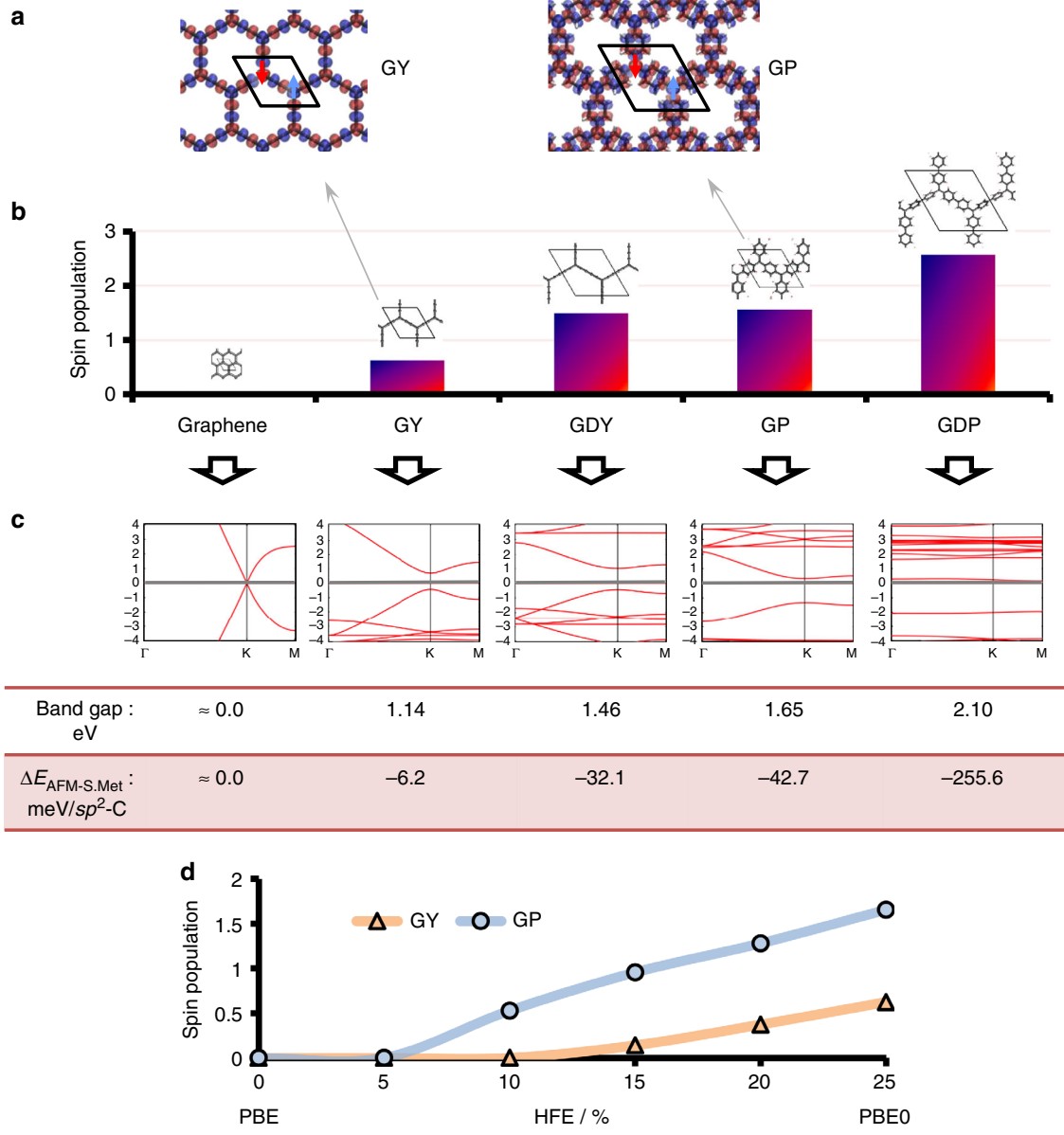

**Fig. 3** Multi-radical AFM state. The results from spin-unrestricted open-shell PBE0 optimizations initialised using an AFM spin configuration. **a** Spin density iso-surface on GY and GP structures (alpha = blue; beta = red), where the unit cell is highlighted. **b** Total absolute spin population (i.e., addition of all atom-partitioned spin-up and spin-down electronic populations) within the unit cell throughout the materials series. **c** Electronic band structures for each case. Below each band structure, we indicate the value of band gaps (eV) opening at the **K** point and the relative energy of the AFM solution with respect to the corresponding semimetallic solution per $sp^2$ triply bonded centre or node of the hexagonal net (each unit cell presents two nodes of the hexagonal net), $\Delta E_{AFM-S.Met}$ in meV. **d** Absolute spin population in number of electrons versus percentage of HFE used in the DFT calculations for GY (orange) and GP (blue)

elusive due to this bias. In this work, we find that PBE0[48] is a functional providing a good balance between localisation and delocalisation (see also Methodology section). Therefore, the main discussion is based on PBE0 results. For comparison with previous works, results obtained by the PBE GGA functional[49] are included in Supplementary Figs. 1–3.

**Semimetallic state**. In Fig. 2, we show the optimised unit cells of each considered material together with the corresponding band structures obtained from spin-unrestricted calculations. In all cases, we find a semimetallic state. Graphene, GY and GDY (Fig. 2a–c, respectively) are one atom thick planar materials and only differ by the distance between triply bonded $sp^2$ carbon atoms siting at the nodes of the hexagonal network, due to the

inclusion of (-C≡C-)$_n$ linkers (where $n = 0$–2), between them. In these cases Dirac cones appear at the high-symmetry reciprocal space **K** point, as previously found[12, 14, 29]. Although the nodal $sp^2$ centres in GP and GDP lay in a plane, these networks cannot be regarded as one-atom planar materials, due to the fact their phenyl rings are partially twisted by ~35° due to inter-ring steric hindrance (see Fig. 2d and 2e, $x$ views). These materials also present a zero band gap feature at **K**. This feature can be understood as resulting from the high symmetry of the hexagonal lattice, which is preserved in all cases.

Overall, it is possible to see a clear reduction in the energy scale of the band structures through the series from graphene to GDP, accompanied with a corresponding flattening of the bands. As reported in Supplementary Fig. 3, this gives rise to a lowering of

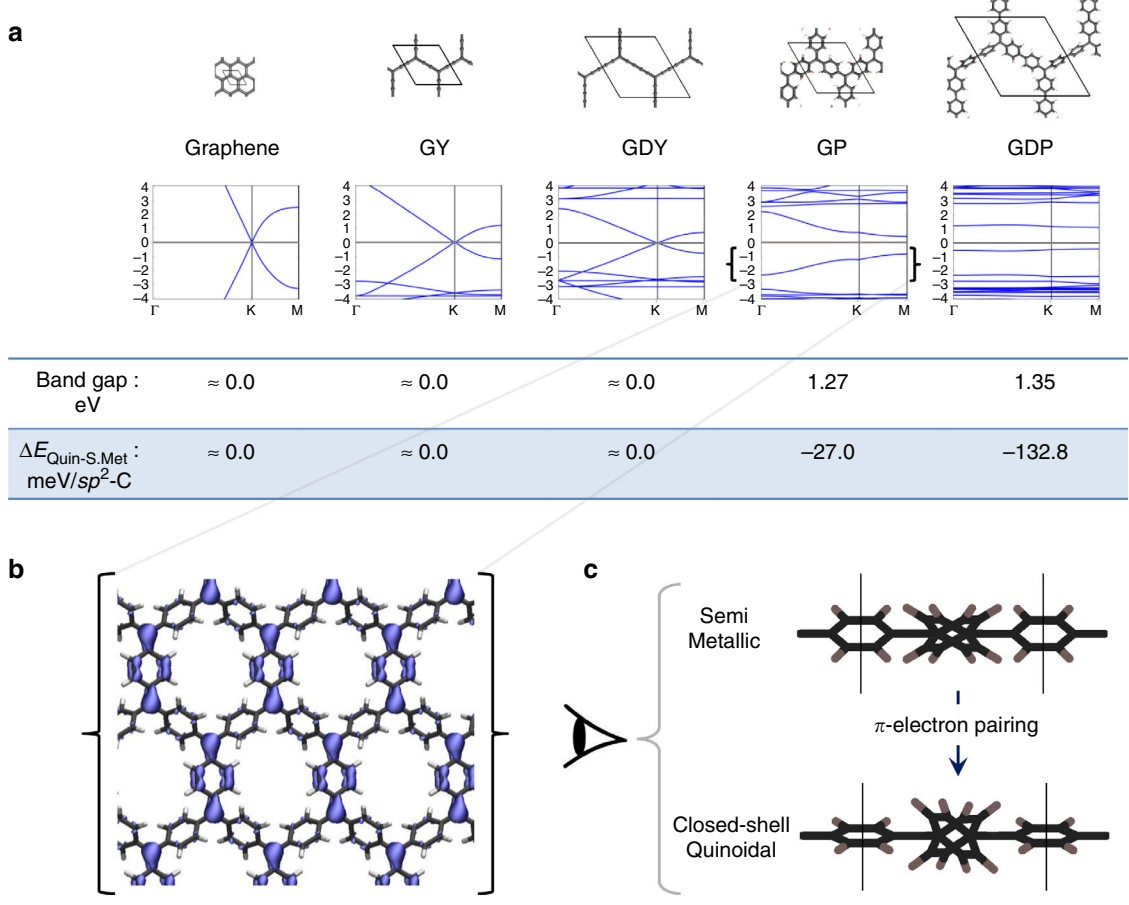

**Fig. 4** Closed-shell quinoidal state. Results from restricted closed-shell PBE0 optimizations. **a** electronic band structures for each material. Below each band structure, we indicate the associated band gaps (eV) and the relative energies of each solution with respect to the semimetallic ones per $sp^2$ carbon centre ($\Delta E_{Quin-S.Met}$) in meV. **b** Iso-surface of the electron density corresponding to the highest occupied crystal orbital at the $\Gamma$ point in GP. **c** x-view of the GP structure in the semimetallic and closed-shell quinoidal solutions, where the planarization of certain phenyl rings upon electron pairing can be observed

the Fermi velocities near the Dirac point going from $5.42 \cdot 10^6$ m/s for graphene to $3.04 \cdot 10^6$ m/s for GDP. These values are in line with those of previous studies[10, 50] and indicate that the increasingly separated $\pi$-conjugated $sp^2$ carbon centres when going from graphene to GDP has only a modest effect on the ballistic transport.

**AFM state**. After confirming the existence of a semimetallic state for each PGOD material, we further investigated possible localised solutions following the Lewis resonance forms depicted in Fig. 1c with an associated reduction of the hexagonal symmetry of each system. In order to examine possible multi-radical AFM states, all materials were optimised employing a spin-unrestricted calculation setup initialised with opposite spin moments on $sp^2$ triply bonded carbon atoms (purple positions in Fig. 1). After optimising the structures, all materials, except graphene, show considerable spin polarisation over the entire network with an anti-ferromagnetic spin alignment over the $sp^2$ triply bonded carbon centres. This spin-polarised solution exhibits a reduced trigonal symmetry. We note that plane wave based DFT calculations also confirm this result (see Supplementary Fig. 5). In Fig. 3a, we show the atom-projected spin density iso-surfaces for GY and GP (see up/down arrows for clarification) as example cases. This clearly demonstrates the partial open-shell multi-radical nature of this series of materials, where every $sp^2$ node bears an AFM coupled localised unpaired electron. In Fig. 3b, we plot the total absolute spin population per material (i.e. the sum

of all atom-partitioned spin-up and spin-down populations in absolute value within each material's unit cell). With the notable exception of graphene, all other materials show a non-negligible spin population, which increases down through the series, towards the maximum value for GDP. The increasing spin population down through the series also suggests that, upon decreasing the $\pi$-overlap that connects neighbouring triply bonded carbon centres, there is a correspondingly higher tendency to stabilise this localised multi-radical state. This chemical rationalisation would also help explain why, in the highly planar and closely linked $sp^2$ network of graphene, such a solution cannot be stabilised. In Fig. 3d it can be seen how the spin density already emerges with low HFE percentages, demonstrating that the presented results are not a spurious singular effect of using PBE0 (25% HFE). Moreover, when no HFE is used, such multi-radical solutions are not found, even when initialising the calculations with an AFM alignment of spin moments. This may explain why such solutions have not been reported in previous works, mainly using pure LDA or GGA functionals.

In Fig. 3c, we show the band structures calculated for each material in the AFM solution using the PBE0 functional. In graphene, where the magnetisation is not exhibited, the Dirac cone is essentially unperturbed. However, for the remainder of the studied materials, with non-negligible spin polarisation, a sizeable band gap opens at the Dirac point, ranging from 1 eV (GY) to 2 eV (GDP). In line with these results, the correlation between radical centres with an AFM alignment of localised spins and the appearance of sizeable band gaps has been previously

**a**

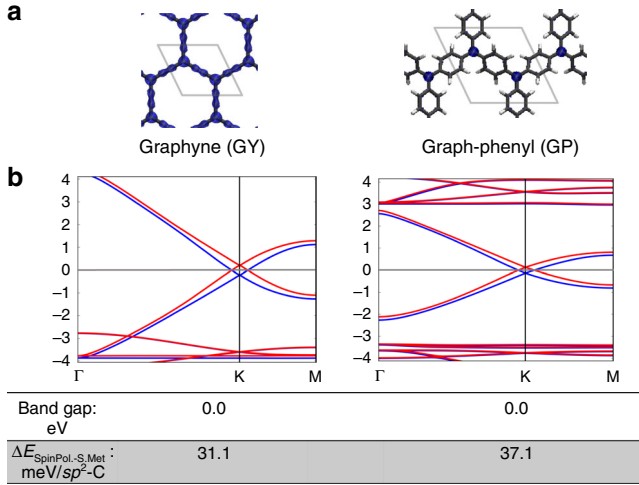

Graphyne (GY)          Graph-phenyl (GP)

**b**

| Band gap: eV | 0.0 | 0.0 |
|---|---|---|
| $\Delta E_{SpinPol.-S.Met}$ : meV/$sp^2$-C | 31.1 | 37.1 |

**Fig. 5** Spin-polarised state. Results from spin-unrestricted open-shell static single point PBE0 calculations forcing a net spin polarisation of 0.15 unpaired electrons using the corresponding previously optimised structures in the semimetallic state. **a** Spin density iso-surface on GY (left) and GP (right) structures (alpha = blue; beta = red), where the unit cell is highlighted. **b** Corresponding electronic band structures for GY (left) and GP (right). Below the plots band gap values (eV) and the energy difference between the spin-polarised state and the corresponding purely semimetallic state ($\Delta E_{SpinPol.-S.Met}$) in meV is given for each material

reported for nanoribbons of graphene[21, 22, 51] and GY[30, 31], as well as for isotropically strained graphene[27]. In all cases the gap opening can be rationalised by the breaking of the hexagonal symmetry of the system. The relative energy per atom with respect to the corresponding semimetallic solutions (i.e., results of Fig. 2) show that such localised AFM multi-radical states (when obtained) are lower in energy as compared to the corresponding semimetallic state for all materials (see Fig. 3c and Supplementary Fig. 6). However, the energy differences in all cases are very small and, hence, both AFM multi-radical and semimetallic states may co-exist under normal conditions.

**Closed-shell quinoidal state**. To find the electronic state corresponding to the closed-shell quinoidal resonant form, following the analogy with Tschitschibabin's hydrocarbon, the structure of each material was optimised using a restricted closed-shell calculation to enforce electron pairing. As it can be seen in Fig. 4a, for graphene, GY and GDY, we were not able to stabilise such a closed-shell quinoidal state and the semimetallic solution prevailed. This indicates that these materials are not prone to have their $sp^2$ carbon π-electrons in localised pairs. We note that other closed-shell solutions based on localised pairing (e.g. Kekulé-like) may be possible for larger cells than used herein.[27] For graphene in particular the difficulty in obtaining gapped states (AFM and/or closed-shell) is highly problematic for its use in electronic applications where band gap control is essential. Consequently, highly invasive methods are needed to open such a band gap in this 2D material, ranging from nanopatterning[18, 19, 52, 53], covalently grafting of atoms and molecules[23, 24, 54] or the application of external strains[27, 28, 55]. Such approaches, however, tend to significantly affect other fundamental and technological interesting properties of graphene (e.g. ballistic charge transport). Conversely, GP and GDP display a sizeable band gap around the Fermi level (see Fig. 4a), which may be indicative of a transition from a delocalised to a localised electronic solution. Band gap values are around 1.3 eV for both GP and GDP, which is a typical value for semiconductor materials.

To better understand the appearance of band gaps in closed-shell GP and GDP, we can examine the iso-surface of the electronic density associated with the highest occupied crystal orbital (see Fig. 4b). Here one observes a quinoidal-like electron distribution, analogous to the closed-shell resonant form in Tschitschibabin's hydrocarbon (Fig. 1c). This demonstrates that the π-conjugated electrons of $sp^2$ centres, previously forming the Dirac cone in the semimetallic state (Fig. 2), or being antiferromagnetically coupled in the multi-radical solution (Fig. 3), now are being locally paired within the periodic 2D structure (Fig. 4b). As for the AFM state, the emergence of these gaps can be understood by the lowering of the hexagonal symmetry; in this case due to the trigonally symmetric charge-ordering resulting from bond formation. GDP exhibits a similar electron density for the highest occupied crystal orbital, as shown in the Supplementary Fig. 7. Examining the structure of GP from the in-plane x-axis view (as represented in Fig. 4c) one can see that, upon moving from the semimetallic solution to the closed-shell quinoidal solution, phenyl rings accomodating the electron pairing become significantly more planar. This twisting mechanism can be understood by the fact that double bonds are being formed between the two in-plane triply bonded carbon nodes, which strongly induce planarity in the bridging phenyl rings (see blue density in 4b). Hence, in GP and GDP, twisting of aryl rings represents an additional structural degree of freedom through which π-electron pairing can be reinforced. In other words GP and GDP have internal structural degrees of freedom which allow for relatively facile symmetry breaking and thus, the emergence of a quinoidal state.

In the perfectly planar GY and GDY, such a mechanism does not exist and, hence, the fully hexagonal symmetric delocalised solution prevails over the quinoidal state. This structure-electronic relationship between aryl ring twist angles and localisation/delocalisation of electrons, previously found for the analogous single-molecules[56], could represent a symmetry breaking tool with which to mechanically induce electron pairing by external strains[57] or, also, to spatially detect paired π-electrons by identifying particularly planarized aryl rings. The quinoidal solution is found to lay 27 meV and 133 meV below the semimetallic solution for GP and GDP, respectively (see Fig. 4a and Supplementary Fig. 6). Therefore, both localised solutions (i.e., AFM multi-radical and closed-shell quinoidal), when viable, are slightly lower in energy than the corresponding delocalised semimetallic state. Although the small energy differences between the three electronic states means that it is probable they would co-exist at finite temperatures, suitable low temperature experiments could potentially selectively stabilise a particular state. Additionally, due to the fact that AFM and closed-shell states have lower symmetries than the corresponding semimetallic state, suitable symmetry breaking influences (e.g. strain, magnetic fields, or lattice vibrations) could enhance the likelyhood of isolating them. Indeed, the fact that semimetallic states require high symmetry could make them difficult to stabilise.

**Spin-polarised state**. Finally, in bi-radical/quinoidal compounds it is known that a ferromagnetic (FM) alignment of unpaired electrons (a triplet state) can be induced by thermal excitations[32, 58]. To assess the feasibility of inducing such a state in our studied PGOD materials, we performed static single point calculations using the semimetallic structures of GY and GP while enforcing an increasing net spin polarisation in the alpha channel. The relative energies of these spin-polarised solutions with respect to the semimetallic solution are reported in Supplementary Fig. 8 for both materials. Figure 5 shows the results when forcing the system to have a net spin polarisation equal to 0.15 unpaired

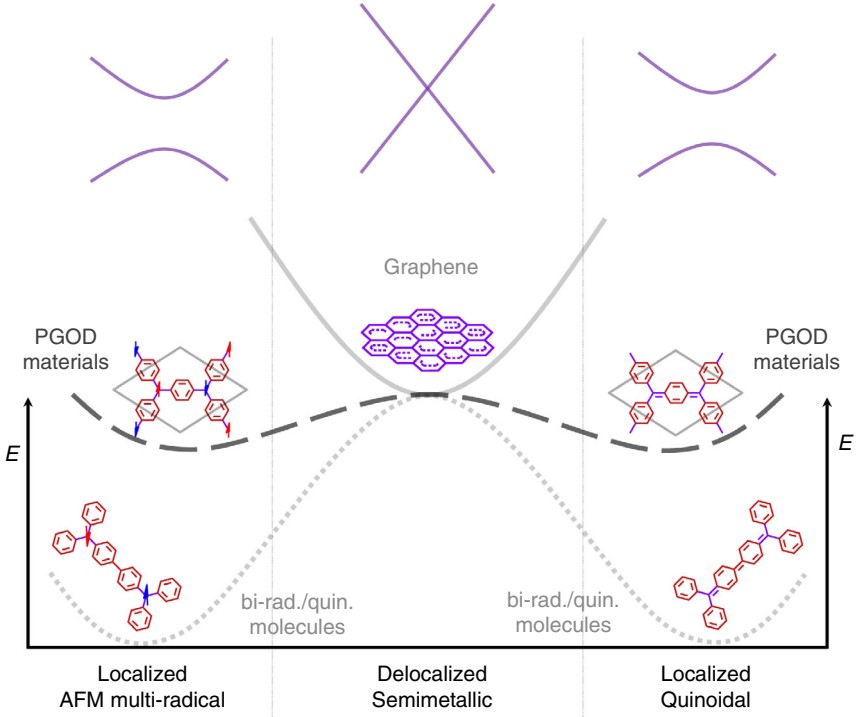

**Fig. 6** PGODs as a bridge between graphene and molecular bi-radicals. PGOD materials (dashed black line) are systems with an electronic character that is intermediate between bi-radical/quinoidal molecules (dotted grey line) and graphene (continuous grey line). Schematic simplified electronic band structures of a generic PGOD material are represented by the solid purple lines above the corresponding electronic states. As such PGODs can exhibit both localised and delocalised solutions at very close energies, and thus possess an electronic tunability not easily obtainable from systems with dominant solutions (e.g., graphene and bi-radical/quinoidal molecules). Note: The relative positions of the different curves have been chosen for ease of illustration and should not be taken to indicate real energy differences between the three different systems

electrons (see Methods section). As it can be seen in Fig. 5a, upon doing so, π-electrons become alpha-spin polarised throughout the network. Examining the GP case, we can see that the spin polarisation (blue density) arises from $sp^2$ triply bonded carbon atoms, in full agreement with the open-shell character of such centres (see above).

As shown in Fig. 5b, these partially spin polarised solutions are similar in character to the semimetallic solutions, as confirmed by the conservation of a spin polarised Dirac cone in the corresponding band structures. Enforcing a spin polarisation excess in the alpha channel induces an occupation of the conduction band in this spin channel (doping with negative charges), emptying the valence band in the opposite one (doping with positive charges) with exactly the same amount. This spin polarisation can also be seen as a breaking of **K**-vector reversal symmetry (time reversal symmetry) resulting in a lifting of the degeneracy of spin-up and spin-down bands. One effect of this spin polarisation is to induce a small (~0.3 eV) energy gap at the previously spin degenerate crossing at **K** resulting in two crossings between bands of the same spin above and below the Fermi level. Additionally, two further crossings between bands of different spin appear at the Fermi level at $\mathbf{K} + \mathbf{k}_1$ and $\mathbf{K} - \mathbf{k}_2$. These two crossings close to **K** are reminiscent of Weyl points, however, as they are constituted by bands of different spin, they indicate the possibility of spin helical Dirac states as found on the edges of graphene nanoribbons (1D)[59] and within surfaces of toplogical insulators (2D)[60]. Applying spin-orbit coupling (SOC) in our calculations reveals that these crossings are maintained and not gapped out (see Supplementary Fig. 9). However, at **K**, the same-spin band crossings above and below the Fermi level are weakly split (0.025 eV) by SOC. The relative energies of these spin-polarised solutions with respect to the purely semimetallic

solutions (see Fig. 5b) demonstrate such a doping process does not cost much energy (31 meV and 17 meV for GY and GP, respectively), suggesting it may be induced by application of moderate magnetic fields or, simply, by thermal excitations, as can occur in molecules exhibiting analogous states[32, 58].

## Discussion

Our work represents a bridge between two important fields in materials science (graphene and PGOD materials) and chemistry (conjugated bi-radical/quinoidal molecules) which, until now, have not had significant overlap. Bi-radical/quinoidal molecules are finite systems and, consequently, their electronic character tends to vary between two localised electronic solutions: namely the bi-radical AFM and the closed-shell quinoidal, as represented by the bottom dotted line in Fig. 6. Because of this bi-stable nature, it is possible to favour one localised solution or the other by chemical design[33, 35], which has raised the interest in such systems for molecular magnetism[34] and molecular electronics[32]. Graphene presents the opposite extreme to bi-radical/quinoidal molecules in this respect. As depicted with the top grey line in Fig. 6, graphene is the perfect semimetal with a single-well energy minimum corresponding to a fully delocalised electronic configuration. This state is a direct consequence of its highly symmetric and planar arrangement of triply bonded $sp^2$ centres. The robustness of this feature has hindered the applicability of graphene for nano-electronics, where the existence of a band gap is a prerequisite[6, 61].

Electronically, the behaviour of PGOD materials lies between that of bi-radical/quinoidal molecules and graphene (see dashed black line in Fig. 6). As we have demonstrated by DFT based electronic structure computations, PGOD materials are a

fascinating system where the fully delocalised semimetallic solution can co-exist with localised multi-radical and closed-shell quinoidal solutions. These solutions can be understood as the result of different symmetry breakings (e.g. via charge-ordering, structural distortion and spin polarisation) from the perfect hexagonal symmetry. As expected from their molecular analogues, in the studied PGOD materials these broken symmetry states are found to be lower in energy than the corresponding symmetric semimetallic states. Moreover, these localised states give rise to band gaps between 1-2 eV, thus representing a clear transition from semimetallic to semiconducting behaviour. Because of the near energetic degeneracy of the semimetallic (and spin-polarised semimetallic) and semiconductor (multi-radical AFM and closed-shell quinoidal), it should be possible to induce particular solutions by external or internal means which deliberately induce symmetry breaking (e.g. strain, application of magnetic or electric fields, chemical design, etc). Hence, PGOD materials present a unique combination of electronic tunability and mechanical robustness, which, we believe, makes them particularly good candidates to become a prominent platform for future nano-electronics.

## Methods

**DFT calculations.** Optimizations of the atomic and electronic structures were performed using different initial electronic guesses to induce different electronic solutions (i.e., semimetallic, closed-shell quinoidal and open-shell anti-ferromagnetic). This set of optimizations was separately performed using the GGA functional: PBE[49] and its hybrid functional extension: PBE0[37, 48]. After each optimisation, single point calculations were performed with the corresponding initial guesses to obtain band structures, Hirshfeld-partitioned atomic spin populations[62], and atom-projected electronic densities. GP and GDP results were obtained from fully symmetrised structures. Each optimisation employed increasingly sized **k**-meshes until convergence of properties was obtained (i.e., total energies and spin densities); here using $6 \times 6 \times 1$ and $18 \times 18 \times 1$ meshes for PBE and PBE0, respectively (see Supplementary Fig. 6). Denser **k**-meshes were required for particularly small unit cells (such as that of graphene) to obtain reasonable band structures. All DFT based calculations were performed employing the all-electron FHI-AIMS code[63], using a light/Tier1 numerical atom-centred orbital (NAO) basis set[64]. This basis set approximately provides results of a similar or higher quality to those obtained with valence triple-zeta plus polarisation Gaussian type orbitals[65]. For the periodic band structure calculations the energy zero (Fermi level) is set by the $G = 0$ component of the long-range electrostatic potential. In addition, the entire set of PBE calculations was also performed using the Vienna ab initio simulation package (VASP)[66], using a plane-wave basis set with a 415 eV kinetic energy cut-off. Due to the very-high computational cost required for hybrid calculations using VASP, only single-point PBE0 calculations using the PBE0-optimised structures from FHI-AIMS were used in our comparisons. We find that in all cases the corresponding PBE and PBE0 results from VASP and FHI-AIMS are fully consistent, thus negating any significant influence of the DFT implementation (e.g. atom-centred numerical versus plane wave basis sets) in our results.

**Use of hybrid functionals.** For capturing the electronic states of mixed-valence organic molecules, some works recommend using hybrid functionals containing 35% HFE, or even higher[45]. For such systems, even widely used hybrid functionals such as PBE0[48] (25% HFE) or B3LYP[67] (20% HFE) still tend to over-delocalise electrons due to the inherent self-interaction error in GGA[68]. However, use of functionals having a relatively large percentage of HFE, although useful for stabilising localised states, can prevent one finding fully delocalised metallic solutions – essential for extended materials such as graphene[69]. As a consequence, we find that the PBE0 functional provides a good balance between electronic localisation and delocalisation for the 2D materials we consider. We note that the optimised structures obtained at using the PBE functional (see Supplementary Fig. 1) are almost indistinguishable to the ones obtained using PBE0 presented in Fig. 2.

**Data availability.** Atomic coordinates and lattice parameters for optimised structures of all considered materials in the semimetallic solution (calculated using the PBE0 functional and a light/Tier1 NAO basis set as implemented in the FHI-AIMS code) are made openly available in Supplementary Data 1.

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

## Acknowledgements

This research was supported by the Spanish MINECO / AEI/FEDER grants CTQ2015-64618-R and CTQ2016-76423-P and, in part, by Generalitat de Catalunya grants 2014SGR97 and XRQTC. We acknowledge access to supercomputer resources as provided through grants from the Red Española de Supercomputación. I.A. acknowledges the Spanish Ministerio de Educación Cultura y Deporte for a FPU PhD scholarship. F.V. acknowledges MINECO for a postdoctoral Ramon y Cajal (RyC) research contract (RYC-2012-10129).

## Author contributions

I.A. and S.T.B.: Came up with the original idea for the study and prepared the manuscript. I.A.: Carried out the FHI-AIMS DFT calculations. F.V.: Calculated the Fermi velocities and carried out the VASP DFT calculations. I.A., S.T.B. and I.P.R.M.: Contributed equally to the interpretation of results. All authors discussed the results and commented on the manuscript. S.T.B.: Supervised the project.

## Additional information

**Competing interests:** The authors declare no competing financial interests.

