## [Peer Review File · Nature Communications]

Reviewers' comments:

Reviewer #1 (Remarks to the Author):

This is a very interesting theoretical paper dealing with various post-graphene organic Dirac (PGOD) structures, which is also very readable for chemists working on molecular level diradicaloids and polyradicaloids. The authors used the PBE0 function and clearly addressed the existence of localized open-shell polyradical and closed-shell quinoidal structures in GP and GDP. They also predicted the significant band gap (semi-conducting) and possible magnetization (small low-spin to high-spin state gap). All these work on large 2D frameworks are nicely consistent to recent studies at molecular levels (e.g. references 34, 66 etc.). This theoretical work is important for experimental development of related materials in future. The paper was well written and I recommend acceptance of this paper in Nature Communications.

Reviewer #2 (Remarks to the Author):

In the manuscript "Existence of multi-radical and closed-shell semiconducting states in post-graphene organic Dirac materials" by Isaac Alc3n et al the authors report about the theoretical investigation of four different post-graphene organic Dirac materials (PGODs), Graph-diphenyl, Graph-phenyl, Graphdiyne and Graphyne in connection to the reference material graphene. In particular they discuss the electronic structure in connection to four possible ground states, given by the semimetallic phase, the closed-shell quinoidal phase, the open-shell antiferromagnetic phase and a ferromagnetic phase. The discussion is based on a chemists point-of-view incorporating detailed descriptions of the different electron bonding characteristics and their influence to the electronic structure.

The applied methods are up-to-date and reasons for the choice of methods are explained in great detail. The provided results are technically sound. The study of Dirac materials and the prediction of real material realizations, especially in 2-dimensional systems has been of ongoing interest during the past decade. The possible existence of Dirac crossings in post-graphene organic Dirac materials (PGODs) is strongly connected to the lattice symmetry and has been discussed various times in various journals. As far as I can judge, the basic message of the paper does not address the existence of the crossings but the mechanisms of establishing a gap in connection to the nature of the chemical bonding in the involved molecules.

Even though, this discussion is novel in the way performed by the authors it could be improved by incorporating aspects of symmetry. With this respect the occurrence of crossings and gaps becomes straightforward. In the semimetallic phase, the discussed PGODs exhibit the same symmetry as graphene together with the same filling-condition (which leads to the semimetallic behavior). The occurring degeneracies at K together with the linearity of the crossing have to occur due to the associated irreducible representations. As reported by the authors Graph-phenyl is the only outlying material showing the linear crossing in the neighborhood of K. This phenomenon is not obvious to me and I could not make sense out of the explanation "it is related to the twisting of the aryl rings". What kind of symmetry is broken by this twisting? Is it possible to compare band structures for a twisted and a non-twisted configuration?

Furthermore, the the closed-shell quinoidal phase, the open-shell antiferromagnetic phase and a ferromagnetic phase all represent symmetry breaking phases. E.g., for the closed-shell quinoidal phase of GP in Fig 4 it looks as charge ordering occurs leading to a trigonal lattice instead of a hexagonal lattice. This should be responsible for lifting the degeneracy at K. I am sure that a similar consideration holds for the antiferromagnetic configuration. The discussion of a ferromagnetic alignment breaks the time-reversal symmetry and with that the degeneracy of spin up and spin down bands. The two crossings close to K should correspond to something like Weyl points. However they will gap out by incorporating spin-orbit coupling.

Additionally, some minor things to improve are:

- i) I recommend to mention the lowering of the total energy for the symmetry breaking phases in comparison to the semimetallic phase more explicitly in the text.
- ii) I don't see the connection of graphene and silicon as mentioned by the authors in the introduction. Probably they refer to the significance of both. However they are different in their underlying properties.
- iii) The last sentence of page 5 "These values, in agreement with [...]" does not make sense to me. Also there is a typo in the sentence before "accompained" should be "accompanied".

Reviewer #1 (Remarks to the Author):

This is a very interesting theoretical paper dealing with various post-graphene organic Dirac (PGOD) structures, which is also very readable for chemists working on molecular level diradicaloids and polyradicaloids. The authors used the PBE0 function and clearly addressed the existence of localized open-shell polyradical and closed-shell quinoidal structures in GP and GDP. They also predicted the significant band gap (semi-conducting) and possible magnetization (small low-spin to high-spin state gap). All these work on large 2D frameworks are nicely consistent to recent studies at molecular levels (e.g. references 34, 66 etc.). This theoretical work is important for experimental development of related materials in future. The paper was well written and I recommend acceptance of this paper in Nature Communications.

Response:

We thank reviewer 1 for his/her positive comments.

Reviewer #2 (Remarks to the Author):

In the manuscript "Existence of multi-radical and closed-shell semiconducting states in post-graphene organic Dirac materials" by Isaac Alc3n et al the authors report about the theoretical investigation of four different post-graphene organic Dirac materials (PGODs), Graph-diphenyl, Graph-phenyl, Graphdiyne and Graphyne in connection to the reference material graphene. In particular they discuss the electronic structure in connection to four possible ground states, given by the semimetallic phase, the closed-shell quinoidal phase, the open-shell antiferromagnetic phase and a ferromagnetic phase. The discussion is based on a chemists point-of-view incorporating detailed descriptions of the different electron bonding characteristics and their influence to the electronic structure.

The applied methods are up-to-date and reasons for the choice of methods are explained in great detail. The provided results are technically sound. The study of Dirac materials and the prediction of real material realizations, especially in 2-dimensional systems has been of ongoing interest during the past decade. The possible existence of Dirac crossings in post-graphene organic Dirac materials (PGODs) is strongly connected to the lattice symmetry and has been discussed various times in various journals. As far as I can judge, the basic message of the paper does not address the existence

of the crossings but the mechanisms of establishing a gap in connection to the nature of the chemical bonding in the involved molecules.

Even though, this discussion is novel in the way performed by the authors it could be improved by incorporating aspects of symmetry. With this respect the occurrence of crossings and gaps becomes straightforward. In the semimetallic phase, the discussed PGODs exhibit the same symmetry as graphene together with the same filling-condition (which leads to the semimetallic behavior). The occurring degeneracies at K together with the linearity of the crossing have to occur due to the associated irreducible representations. As reported by the authors Graph-phenyl is the only outlying material showing the linear crossing in the neighborhood of K. This phenomenon is not obvious to me and I could not make sense out of the explanation "it is related to the twisting of the aryl rings". What kind of symmetry is broken by this twisting? Is it possible to compare band structures for a twisted and a non-twisted configuration? Furthermore, the closed-shell quinoidal phase, the open-shell antiferromagnetic phase and a ferromagnetic phase all represent symmetry breaking phases. E.g., for the closed-shell quinoidal phase of GP in Fig 4 it looks as charge ordering occurs leading to a trigonal lattice instead of a hexagonal lattice. This should be responsible for lifting the degeneracy at K. I am sure that a similar consideration holds for the antiferromagnetic configuration. The discussion of a ferromagnetic alignment breaks the time-reversal symmetry and with that the degeneracy of spin up and spin down bands. The two crossings close to K should correspond to something like Weyl points. However they will gap out by incorporating spin-orbit coupling.

Response:

We fully agree that symmetry arguments enable one to establish the existence of gaps and crossing from a general top-down perspective, as clearly summarised by the reviewer, and as powerfully used many times in the literature. The main focus of our work, as the reviewer also notes, is to provide a complementary bottom-up chemical rationalisation for the low energy states in PGODs. In particular we link the experimentally demonstrated properties of radical molecules with the extended states in analogous PGODs. In this way we lead to a chemical/molecular understanding of how low energy quinoidal (gapped), multi-radical anti-ferromagnetic (gapped) and ferromagnetic spin-polarized (ungapped) states in such PGODs emerge from the dominant graphene-like semimetallic phase. Following the reviewers recommendation we now include some associated discussion of the role of symmetry in the origin of these states which indeed is likely to enhance the general understanding of our findings. Specifically, we now include:

(i) A sentence noting that PGODs having the same hexagonal symmetry as graphene exhibit a semi-metallic phase.

(ii) A comment describing how the bond formation (charge ordering) in the closed-shell quinoidal phase and the spin-ordering in the antiferromagnetic phase leads to a breaking of the hexagonal symmetry of the lattice leading to a trigonal lattice where the degeneracy at K is lifted (i.e. a gap is opened).

(iii) Regarding the ferromagnetic phase, we now note that the breaking of k-vector reversal symmetry (time reversal symmetry) results in a lifting of the degeneracy of spin-up and spin-down bands. As the reviewer points out, the resulting two crossings close to K are reminiscent of Weyl points, however, in this case, the crossing bands are of different spin. As such they indicate the

possibility of helical Dirac states as found on the surfaces of topological insulators (2D) and on the edges of graphene nanoribbons (1D). Applying spin orbit coupling (SOC) in our calculations reveals that these crossings are maintained and not gapped out. However, at K the same-spin crossings above and below the Fermi level are weakly split by SOC. The results of these calculations incorporating SOC are now included in the Supp. Info. An associated discussion of these results and two references to papers discussing helical Dirac states have also been added to the main text.

(iv) The reviewer notes that Graph-phenyl (GP) has linear crossings in the neighbourhood of K indicating a symmetry breaking. The slight ring-twisting we refer to in the text indeed can be interpreted as a breaking of the hexagonal symmetry of the lattice. Applying hexagonal symmetry to our calculations we find that, as expected, the crossing occurs at K. The resulting symmetric structure also has a very similar energetic stability to that of the broken symmetry structure. For simplicity of exposition we now use the symmetric version of Graph-phenyl in the main text and include the results relating to the broken symmetry structure in the Supp. Info.

Response to minor points:

i) I recommend to mention the lowering of the total energy for the symmetry breaking phases in comparison to the semimetallic phase more explicitly in the text.

We agree that this is an important message of our work and we have now more explicitly emphasised this result.

ii) I don't see the connection of graphene and silicon as mentioned by the authors in the introduction. Probably they refer to the significance of both. However they are different in their underlying properties.

Although they are materials with different inherent electrical properties (silicon: semiconductor, graphene: semimetal), the potential use of graphene for nano-transistors has become an important line of research (see: Schwierz, F. Graphene transistors. *Nat. Nanotechnol.* **5**, 487 (2010)). Indeed, graphene has been promoted by many as the natural replacement of silicon for future nano-electronics' applications. It is in this context that we mention silicon and graphene.

iii) The last sentence of page 5 "These values, in agreement with [...]" does not make sense to me. Also there is a typo in the sentence before "accompanied" should be "accompanying".

We have tried to clarify the description of the conclusions summarised in the sentence on page 5. The typo is corrected.

REVIEWERS' COMMENTS:

Reviewer #2 (Remarks to the Author):

The authors responded to all questions addressed within my report. Furthermore, they improved the content towards including aspects of symmetry breaking. I think including these changes, the paper is a significant contribution in the field.